# The bidirectional association between depressive symptoms, assessed by the HADS, and albuminuria–A longitudinal population-based cohort study with repeated measures from the HUNT2 and HUNT3 Study

Lise Tuset Gustad[1,2,3©]*, Anna Marie Holand[4,5©], Torfinn Hynnekleiv[6], Ottar Bjerkeset[1,7], Michael Berk[8,9], Solfrid Romundstad[3,10]

1 Faculty of Nursing and Health Sciences, Nord University, Levanger, Norway, 2 Department of Circulation and Medical Imaging, Faculty of Medicine and Health, Norwegian University of Science and Technology (NTNU), Trondheim, Norway, 3 Department of Medicine, Nord-Trøndelag Hospital Trust, Levanger Hospital, Levanger, Norway, 4 Faculty of Education and Arts, Nord University, Levanger, Norway, 5 Department of Public Health and Nursing, Faculty of Medicine and Health Sciences, Norwegian University of Science and Technology (NTNU), Trondheim, Norway, 6 Division of Mental Health, Department of Acute Psychiatry and Psychosis Treatment, Innlandet Hospital Trust, Reinsvoll, Norway, 7 Department of Mental Health Sciences, Faculty of Medicine and Health, Norwegian University of Science and Technology (NTNU), Trondheim, Norway, 8 IMPACT–the Institute for Mental and Physical Health and Clinical Translation, School of Medicine, Barwon Health, Deakin University, Geelong, Australia, 9 Orygen, The National Centre of Excellence in Youth Mental Health, Centre for Youth Mental Health, Florey Institute for Neuroscience and Mental Health and the Department of Psychiatry, The University of Melbourne, Melbourne, Australia, 10 Department of Clinical and Molecular Medicine, Faculty of Medicine and Health, NTNU, Trondheim, Norway

© These authors contributed equally to this work.
* lise.t.gustad@nord.no

## Abstract

### Background

Both albuminuria and depression are associated with cardiovascular disease, reflecting low-grade systemic inflammation and endothelial dysfunction. They share risk factors including weight, blood pressure, smoking and blood glucose levels. This longitudinal study aimed to examine bidirectional associations between depression symptoms, indexed by the Hospital Anxiety and Depression scale (HADS), and the inflammation marker albuminuria.

### Methods

2909 persons provided urine samples in both the second (HUNT2, 1995–97) and third wave (HUNT3, 2006–2008) of the Trøndelag Health Survey, Norway. We used a generalized linear regression model (GLM) and ANOVA to assess the association between albuminuria levels (exposure HUNT2) with depression symptoms (outcome in HUNT3); and between depression symptoms (exposure HUNT2) with albuminuria (outcome HUNT3). Depression symptoms were measured with the HADS Depression Scale, analyzed utilising the full 7 items version and analyses restricted to the first 4 items (HADS-D and HADS-4). We accounted for confounders including baseline individual levels of the exposure variables.

**Data Availability Statement:** Due to confidentiality HUNT Research Centre wants to limit storage of data outside HUNT databank, and researchers need approval by the Regional Ethical Committee (rek-midt@mh.ntnu.no) for handling of HUNT data files. HUNT Research Centre has precise information on all data exported to different projects and there are no restrictions regarding data export given approval of applications to HUNT Research Centre (http://www.ntnu.edu/hunt/data). Interested readers may send e-mail to kontakt@hunt.ntnu.no for more information. We can confirm that interested readers will be able to access the data in the same manner that the authors have accessed it if they have a legitimate reason to be considered by the HUNT Research Centre and the Regional Ethical Committee.

**Funding:** LTG is funded by the Liaison Committee for education, research, and innovation in Central Norway. The funding body had no role in the designs of the study, collection, analysis, interpretation of data, and in writing the manuscript.

**Competing interests:** MB has received Grant/Research Support from the NIH, Cooperative Research Centre, Simons Autism Foundation, Cancer Council of Victoria, Stanley Medical Research Foundation, Medical Benefits Fund, National Health and Medical Research Council, Medical Research Futures Fund, Beyond Blue, Rotary Health, A2 milk company, Meat and Livestock Board, Woolworths, Avant and the Harry Windsor Foundation, has been a speaker for Astra Zeneca, Lundbeck, Merck, Pfizer, and served as a consultant to Allergan, Astra Zeneca, Bioadvantex, Bionomics, Collaborative Medicinal Development, Lundbeck Merck, Pfizer and Servier – all unrelated to this work. The COIs does not put a restriction on PLOS ONEs policies. See Availability of data and materials for details on how to access the data.

## Results

In this 10-years follow-up study, we found no statistical evidence for an association between baseline depression symptoms and subsequent albuminuria, nor between baseline albuminuria and subsequent depression symptoms. For albuminuria, only 0.04% was explained by prior depression, and for depression, only 0.007% was explained by previous albuminuria levels. The results were essentially the same for the shorter HADS-4 measure.

## Conclusion

There does not appear to be a longitudinal association between albuminuria and depression measured by the HADS.

## Introduction

Depression symptoms may be more prevalent in people with albuminuria—leading to hypotheses regarding causal directions. In a previous cross-sectional population-based cohort study, albuminuria was associated with depressive symptoms and depressive episodes, even at levels of albuminuria that do not fulfil the chronic kidney disease (CKD) criteria [1]. This led to a hypothesis that albuminuria could serve as a marker for individuals at risk of depression [1]. Others have put forward a hypothesis that depression symptoms might be risk factors for albuminuria, or that the observed overlap is explained by other participant traits, i.e. reflect confounding [2]. One cross-sectional study found that the association between HADS-assessed depression and albuminuria was explained mainly by age and comorbidity [3]. Nevertheless, these conflicting cross-sectional data that suggest potential bidirectional causal relationships between albuminuria and depression symptoms might inform detection or prevention strategies for future cardiovascular disease (CVD). These speculations are important to explore as albuminuria and depressive symptoms both are potentially preventable risk factors for CVD [4–8].

To date, one large-scale study has investigated the longitudinal bidirectional relation between depression and albuminuria [9]. Regarding the causal direction from urinary albumin to risk of depression, Liu et al (2022) found no clear evidence of elevated urine albumin associated with future depression, while reduction of eGFR was associated with increased risk of future depression. Further, one small-scale longitudinal study of 16 children assessed the relationship between proteinuria and perceived stress [10]. While proteinuria and albuminuria are not the same, large amounts of albuminuria approximate a positive proteinuria dipstick [11]. However self-reported proteinuria utilising albustix, as used by Bakkum et al (2019) [10], is a method more prone to user error than laboratory-confirmed proteinuria [12]. In examining the causal relationship from depression to albuminuria, none of the above longitudinal studies included such analysis.

We therefore aimed to examine if levels of albuminuria were associated with future depression symptoms and if depression symptoms were associated with future levels of albuminuria. These bidirectional hypotheses were explored in a longitudinal population-based follow-up study of the general population, applying adjustments for important confounding factors such as age, weight, blood pressure, smoking and blood glucose levels, that is associated with both albuminuria and depression [5, 13–18]

## Methods

### Study population

This study is based on participants who provided urine samples and answered questions regarding depression symptoms in both the second HUNT wave (HUNT2, 1995–97) and the third HUNT wave (HUNT3, 2006–2008) [6, 19, 20].

### The albuminuria subpopulation

HUNT2 invited 11501 of the 65003 (17.7%) consenting participants to provide urine samples (the HUNT2 Albuminuria substudy). 6779 (58.9%) of the participants reported diabetes or the use of hypertensive medications, and 5% were randomly selected (n = 2686) [21]. While 9135 were invited to provide urine samples in HUNT 3, 4260 (46.6%) of these previously provided urine samples in HUNT2; 2768 (40.8%) of those with diabetes and hypertension, and 1492 (55.5%) of the random selection [6]. Consent to participate was given twice by 2909 of the re-invited (68.2%); 1843 (63.4%) of the diabetic and blood pressure samples, and 1066 (36.6%) of the randomly selected (see Fig 1 for the flow chart regarding inclusion).

### Measures

**Laboratory data (blood and urine).** Blood sampling was performed in a non-fasting state at the clinical examination. HUNT participants returned the urine samples in standardized receptacles using prepaid envelopes. Fresh blood and urine samples were analyzed at an accredited laboratory (ISO-9001 certified and ISO/IEC-17025) in Levanger Hospital (Norway). For the analysis, HUNT2 used a Hitachi 911 autoanalyzer (Hitachi, Mito, Japan) with reagents from Boehringer Mannheim (Mannheim, Germany), and HUNT3 used an Architect ci8200 autoanalyzer (Abbot Diagnostic, Longford, Ireland) [20].

| HUNT2 (1995-97) | |
|---|---|
| Invited >20 years | 93898 |
| Participated, 69,2% | 65003 |
| Invited Albuminuria substudy, 17.7% | 11501 |
| Participated Albuminuria substudy, 84.8% | 9752 |
| **Selection criteria participants albuminuria** | |
| Diabetes or blood pressure medication, 58.9% | 6779 |
| Random 5% selection, 49.1% | 2686 |
| Other, 2.9% | 287 |

| HUNT3 (2006-08) | |
|---|---|
| Invited >20 years | 93860 |
| Participated, 53.8% | 50556 |
| Invited Albuminuria substudy, 18.0% | 9135 |
| (of those reinvited after HUNT2) | 4260 |
| Participated Albuminuria substudy, 64.2% | 5861 |
| **Selection criteria participants albuminuria** | |
| Diabetes in HUNT3, 12.5% | 738 |
| Random 5% selection, 33.9% | 1985 |
| Other, 3.2% | 188 |
| HUNT2 follow-up sample, 50.3 % | 2909 |
| Details HUNT2 follow-up sample, n= 2909 | |
| Diabetes or blood pressure medication, 63.4% | 1843 |
| Random 5% selection, 36.6% | 1066 |

**Fig 1. Flow chart of recruitment of the participants in the study.**

HUNT used the Jaffe method, calibrated to isotope-dilution mass-spectroscopy, to measure the serum concentration of creatinine (Se-Crea). Calibrated se-Crea was used to calculate estimated glomerulus filtration rate (eGFR), using the Chronic Kidney Disease Epidemiology Collaboration equation (CKD-EPI) [22]. Immunoturbidimetric methods using antihuman serum albumin were applied to determine urine albumin and calculate Albumin Creatinine Ratio (ACR). In HUNT2, the supplier was Dako AS, Glostrup, Denmark [23], and in HUNT3 Abbot Laboratories [20]. Detailed information on quality control and calibration methods have been previously published [24]. A correction equation was applied to ensure identical calibration of the methods: u-albuminHUNT-3$_{Hunt-2}$ level = [(0.81_u-albumin HUNT-3)0.5] [6].

**Clinical examination.** Trained nurses performed standardized clinical examinations at baseline [20]. The assessments included measurements of the participants' height and weight (both with light clothing without shoes, to the nearest cm or kg). Body mass index (BMI) was calculated as kg/m$^2$. The average of the second and third systolic blood pressure (SBP) measurements was used, recorded by an automatic oscillometric method (Dinamap 845XT; Criticon, Tampa, Florida, USA) after > 5 minutes resting in a sitting position.

**Self-report.** Participants self-reported depression symptoms experienced in the past week using the depression subscale of the Hospital Anxiety and Depression scale (HADS). The HADS depression (HADS-D) subscale includes seven items that screen for non-somatic depression symptoms like anhedonia and psychomotor retardation [25]. The HADS-D measures the severity of depression in the general population with a sensitivity and specificity of ~80% [26]. Each question is answered using a scale of 0–3, i.e., the HADS-D subscale ranges from 0 to 21. In addition to the seven-item HADS-D, we also used only the sum of the first four HADS-D items (HADS-4), which has shown almost equal sensitivity and specificity as the seven-item version (HADS-7), however it may yield less heterogeneity and more power [27]. HADS-4 has statistical power of 0.76, while the HADS-D has power of 0.37 due to a more homogenous depression phenotype. The HADS-4 ranges from 0–12 and has almost equal sensitivity and specificity for depression as the HADS-D.

In HUNT 3, the participants self-reported diabetes (yes/no) and/or using hypertensive medication (yes/no). The HUNT2 response of hypertension was re-coded into yes/no by merging the "present use of blood pressure medication" and "previous use of blood pressure medication" into the yes-category. Weekly alcohol consumption was based on participants' reports of how many beers, wine, and liquor units they usually consumed in two weeks. The amount of beer, wine, and liquor units were combined and categorized; zero units per week, or answering yes to being an abstainer, was defined as "abstainer," 1–14 units as "light drinker," 15–28 units as "moderate drinker" and more than 28 units as "heavy drinker". We constructed a variable for "education and work" based on self-reported education (for HUNT2 participants) and the three-class version of the Classification of Occupation Codes from Statistics Norway (for HUNT3 participants). Category 1 included participants if they had less than ten years of education or belonged to the working class; category 2 included participants reporting 10–12 years of education or belonged to the intermediate working class; category three was applied when participants had >12 years of education or belonged to the salaried class. Smoking was self-reported as "never, previous and current smoking" [20].

**Ethics approval and consent to participate.** All participants in HUNT2 and HUNT3 voluntarily presented for examination and gave written and informed consent to use their data for research. All HUNT participants have the ability to withdraw from the registry. This study was approved by the Regional Committee for Medical and Health Research Ethics in South East Norway (2010/178-2/REC Mid-Norway) with the last extension in 2019 (ref no 12364). The study is also approved by the Data Access Committee (DAC) for Nord-Trøndelag Hospital Trust and the DAC from the HUNT Study.

## Statistical analysis

Repeated measurements from all participants attending HUNT2 and HUNT3 were used. Per protocol, we replaced 1–2 missing items on the HADS depression subscale with 6/7 and 5/7 of the values provided, and defined these as our complete case [28]. No such replacements were performed on missing items for the HADS-4. As the laboratory quality program detected three months (January–March 2008) with false elevated ACR values, we re-coded these values as missing [6]. If the participants self-reported a urinary tract infection during the last week, experienced persistent hematuria over the previous year, and if women were pregnant or menstruating at collection time, the urinary measurements were also listed as missing

In the descriptive analysis, we evaluated depression symptoms, albuminuria, and covariates, for both study waves, with HUNT2 being defined as baseline and HUNT3 as the end-point. Exposure of depression symptoms was defined as HADS-D levels in HUNT2 and outcome as HADS-D levels in HUNT3. Albuminuria was defined as an exposure in HUNT2 and an outcome in HUNT3.

To explore the association between depression symptoms and albuminuria, we performed a Spearman Rho correlation between the exposure variables, as the assumptions for a Pearson correlation were not met. Bidirectionality in our study is indicated if the exposure to one variable in HUNT2 is associated with change in the other variable in HUNT3 (the outcome). The two hypotheses (H) explored were thus;

H1: Depression symptoms in HUNT2 (exposure) are associated with albuminuria levels in HUNT3 (outcome).

H2: Albuminuria levels (exposure) in HUNT2 are associated with depression symptoms in HUNT3 (outcome).

We considered each hypothesis separately using a generalized linear regression model (GLM), adjusting for confounding factors *(k)* at HUNT2. See S1 File for an overview of the two statistical formulas used, equation 1 (Eqn1) and equation 2 (Eqn2), one for each hypothesis.

The confounding (*k*) factors were added in three different models. Model 1 included *k* = age and sex. Model 2 adjusted for the baseline level of the outcome variable ($y_i$ or $z_i$), as repeated measures take height for the within-individual association for paired measurements. Model 3 added further adjustments for *K* = education, BMI, smoking, cholesterol, eGFR, diabetes, SBP, antihypertensive medication, and alcohol in addition to model 2.

Regression models were first fitted to the complete dataset (including missing) using model 3. For Eqn.1, the HADS-scores were treated as a continuous (range 0–21 for HADS-7 and 0–12 for HADS-4), and a GLM with a gamma-distributed dependent variable, and an identity link, was fitted. A value of 1 was added to the score, as gamma distribution does not exist for 0, and HADS scores include the value 0. When transforming back to the original scale, the expectation (E) of model 1–3 is $E(y+1) = E(y) + 1$. Therefore, the only transformation is for the intercept (b0–1). This should avoid using a log transformation that can lead to bias and non-linearity in the interpretation of the back-transformed estimates. For Eqn.1, a pseudo $R^2$ was calculated as 1-(residual deviance/null deviance) and p-value is calculated from a deviance goodness of fit test [29]. $R^2$ is the coefficient of determination.

For Eqn.2; the albuminuria variable was also treated as a continuous variable, and was log-transformed and fitted with a linear model with a Gaussian distribution. The estimates were then back-transformed using the exponential function and reported as a percentage.

A visual inspection of the residual plots revealed no obvious deviations from the assumptions of the linear regression models. Further, the variance inflation factor (VIF) was smaller

than 5 for all predictors (most predictors have a VIF of 1) and did not indicate any multicollinearity causing concern in the dataset.

The missing values ranged between 0 and 9%. We assumed missing at random (MAR) for the included variables. MAR was tested by checking if missing values depended on age, sex, education, or comorbidities. Missing data were imputed using multiple imputations (MI) by fully conditional specification (FCS) allowing a flexible assumption for each variable [30]. The MI routine generated a total of 20 complete datasets by 20 iterations in the "MI chained"-routine. Passive imputation was performed for the log transformed response of albuminuria. All variables used in model 3 were included for the imputation procedure, and measurements from both study waves were used for imputation. Analytical results from MI were compared to results from the complete case. Visual inspection of the imputation indicated an appropriate imputation. The regression models were fitted to the complete dataset (with missing) to find an appropriate model, and further fitted to the imputed dataset to improve the power. Analysis of variance (ANOVA) was used to assess the overall fit of the models and find the proportion of variance in the outcome, explained by exposure (effect size of variables in model 3).

Sensitivity analysis was conducted to investigate the plausibility of the MAR assumption for HADS. The delta (δ) adjustment technique (described in paragraphs 3.8 and 9.2 in [30]) was used to explore the effect of increasing the imputed HADS scores by adding an amount δ to the values imputed under MAR. The chosen δ-values for the HADS scores from HUNT2 to HUNT3 were 0 (MAR) and 1, 2, 3, 4 and 5 (Missing Not At Random assumption). As 17.6% of the albuminuria values in HUNT3 were set to missing due to measurement error in a 3-month period at the lab, we know that these values are missing under MAR assumptions. The rest of the missingness in albuminuria (2.2% in HUNT2 and 9% in HUNT3) was related to sex, and the effect of sex should be accounted for in the models.

A potential interaction effect between age and sex on exposure was explored in model 1 and 3. This interaction effect was only found to be significant for model 1 for Eqn 2. This effect was however considered small and the interaction term did not significantly improve the other models (see S1 Table). The interaction effect was thus chosen to be excluded from the models as this interaction effect uses degrees of freedom and complicates the model.

Descriptive statistics was performed using Stata Statistical Software, TX: StataCorp LLC 2017. All other analyses were performed using R (R Core Team 2021). The MI method was performed in R using the *mice* package v3.13.0 [31].

## Results

Table 1 shows the descriptive statistics for the 2909 participants that attended two study waves. The participants had higher albuminuria levels in HUNT 3 than in HUNT2, as expected with ten years of ageing. The average HADS-D score was stable across decades.

The pooled correlation between HADS depression symptoms and albuminuria levels in HUNT2, and between each variable measured in HUNT2 with the variable measured in HUNT3, is illustrated in Table 2. The results indicate a non-significant association between depression symptoms at baseline and albuminuria levels at HUNT3 (r = .04, p = .12). A negligible association was found between albuminuria levels at baseline and HADS depression scores in HUNT3 (r = .07, p < .001 for HADS-D and r = 0.05, p<0.01 for HADS-D 4 items), although these associations were significant. The association between repeated depression symptoms across the two study waves was moderately positive (r = .54, p < .001 for HADS-D and r = .50, p<0.001 for HADS-D 4 items), and the association between repeated albuminuria levels across the two studies was weakly positive (r = .26, p < .001).

**Table 1.  Baseline characteristics of the participants at HUNT2 (1995–97) and HUNT3 (2006–2008), n = 2909.**

| Variable | HUNT2 | m, n (%) | HUNT3 | m, n (%) |
|---|---|---|---|---|
| Sex, male, n (%) | 1357 (46.6) | 0 (0) | NA | NA |
| Age, years, mean (SD) | 54.5 (11.5) | 0 (0) | 65.6 (11.5) | 0 (0) |
| ACR, mg/mmol, mean (SD) | 1.2 (2.2) | 98 (1.5) | 2.7 (7.8) | 262 (9.0)* |
| Depression, HADS-D score, mean (SD) | 3.7 (3.1) | 45 (1.6) | 3.8 (3.0) | 193 (6.6) |
| Body Mass Index, kg/m2, mean (SD) | 27.7 (4.3) | 0 (0) | 28.7 (4.6) | 13 (0.4) |
| eGFR (CKD-epi), mean (SD) | 81.0 (15.8) | 0 (0) | 77.2 (16.8) | 27 (2.3) |
| Cholesterol, mg/mmol, mean (SD) | 6.1 (1.2) | 0 (0) | 5.4 (1.1) | 58 (1.9) |
| Systolic blood pressure, mmHg, mean (SD) | 144.2 (20.7) | 0 (0) | 139.2 (20.3) | 5 (0–2) |
| Self-reported diabetes, yes, n (%) | 313 (10.6) | 0 (0) | 525 (18.0) | 3 (0.1) |
| Self-reported CVD, yes (n %) | 337 (11.6) | 0 (0) | 622 (21.4) | 0 (0) |
| Self-reported blood-pressure medication, yes, n (%) | 1753 (60.2) | 0 (0) | 2043 (70.2) | 0 (0) |
| Smoking Status, yes, n (%) | | 21 (2.1) | | 94 (3.2) |
| Never | 1301 (44.7) | | 1162 (39.9) | |
| Previous | 1059 (36.4) | | 1326 (45.6) | |
| Current | 528 (18.2) | | 327 (11.2) | |
| Alcohol Status, units per week, n (%) | | 98 (4.7) | | 32 (1.1) |
| <1 | 1158 (39.9) | | 793 (27.3) | |
| 1–7 | 1319 (45.3) | | 1448 (49.8) | |
| 8–14 | 261 (8.8) | | 505 (17.4) | |
| >15 | 73 (2.3) | | 131 (4.5) | |
| Education (HUNT2) or work status (HUNT3), n (%) | | 96 (3.3) | | 193 (6.6) |
| Ten years of school or less or unskilled worker | 1235 (42.5) | | 734 (25.2) | |
| 10–12 years of school or intermediate working class | 1074 (36.9) | | 1619 (55.7) | |
| >12 years of School or salariat working class | 504 (17.3) | | 363 (12.5) | |

Abbreviations: m = missing; n = number of participants; ACR = Albumin Creatinine ratio in urine, CVD = Cardiovascular disease.

*Additional 514 (17.6%) were set to missing due to deviations in urinary analysis, or self-report of urinary tract infection during the last week, persistent hematuria over the previous year, and whether women were pregnant or menstruating at collection time.

The results for the GLM related to the effects of albuminuria in HUNT2 on depression symptoms in HUNT3 (eqn 1) are given in Table 3. Table 3 shows that for every one-unit increase in albuminuria values in HUNT2, the average depression symptom score in HUNT3 decreases by 0.001 points for HADS-D-7 and decreases by 0.003 points for HADS-D-4. The confidence intervals were wide and inconclusive for both HADS-depression measures. However, depression levels in HUNT2 were associated with depression levels in HUNT 3, ($\beta$ = .53 (.49, .57), p < .001). All the covariates in model 3 poorly accounted for the proportion of variation, pseudo-R2 = 0.29.

Table 4 indicates that for every one-unit additional HADS-score in HUNT2, albuminuria in HUNT3 increases by 0.5% for the 7-item HADS and decreases by 0.4% for the 4- item HADS. Both estimates have wide and inconclusive confidence intervals. Albuminuria in HUNT2 was also associated with albuminuria in HUNT3, and the HUNT2 albuminuria levels explained 4.08% of the albuminuria levels in HUNT3. All the covariates in model 3 poorly accounted for the proportion of variation, pseudo-R2 = 0.15.

In S1 Table, the results from complete case analysis for Eqn1 and Eqn2 are shown with the interaction term between sex and age. In S2 Table, the results in complete case analysis are shown without the interaction term between sex and age. The complete case analysis showed no significant changes from the multiple imputed data.

**Table 2. Spearman correlation between the depression symptoms and albuminuria[*].**

| Correlation | | HADS-D 7 items[1] | Albuminuria [2] | HADS-D 7 items[1] | Albuminuria[2] |
|---|---|---|---|---|---|
| | | HUNT 2 | | HUNT 3 | |
| Depression | HUNT 2 | 1 | | | |
| Albuminuria | | r = .02 (.02), p = .20 | 1 | | |
| Depression | HUNT 3 | r = .54 (.01), p < .001 | r = .07 (.02), p < .001 | 1 | |
| Albuminuria | | r = .04 (.02), p = .12 | r = .26 (.02), p < .001 | r = .07 (.02), p < .001 | 1 |
| | | HADS-D 4 items[3] | Albuminuria[2] | HADS-D 4 items[3] | Albuminuria[2] |
| Depression | HUNT2 | 1 | | | |
| Albuminuria | | r = .008 (.02), p = .69 | 1 | | |
| Depression | HUNT3 | r = .50 (.02), p < .001 | r = .05 (.02), p < .01 | 1 | |
| Albuminuria | | r = .03 (.02), p = .18 | r = .25 (.02), p < .001 | r = .06 (.02), p < .01 | 1 |

[*]Results from imputed dataset, Degrees of freedom = 2907.

Abbreviations: r = correlation coefficient, () = (standard error of pooled estimates), p = p-value.

[1]HADS-D 7 items = Hospital Anxiety and Depression Scale-Depression subscale, all seven items on the scale. Self-reported depression symptoms (0–21)

[2]Albuminuria is measured by Albumin Creatinine Ratio (ACR) in urinary samples.

[3]HADS-D 4 items = Hospital Anxiety and Depression Scale-Depression subscale, the first 4 items of the HADS-D 7 items scale. Self-reported depression symptoms (0–12)

In S3 Table the ANOVA analysis for albuminuria levels in HUNT3 and depression symptoms in HUNT3 are shown as outcomes respectively. For the ANOVA analysis, HADS-D 7 item and HADS-D 4 item were comparable and the results for HADS-D 7 item only are shown. Comparing the fit of the nested models using ANOVA (F-test) indicated that model 3 is more accurate than both model 1 and model 2 for both outcomes. Regarding results from the ANOVA for albuminuria in HUNT3, the etas confirm the results of the $R2$ values presented earlier. The ANOVA shows that only a small proportion of the variance is associated with the exposures. For albuminuria levels in HUNT3, 4% (Partial eta$^2$ = 0.04) of the variance is explained by previous albuminuria values in HUNT2, whilst only 0.04% is explained by depression levels in HUNT2. For the depression symptoms in HUNT3, the ANOVA table indicates that very little of the variance is associated with albuminuria in HUNT2 (0.007%), whilst depression symptoms in HUNT3 have a moderate association with depression symptoms in HUNT2 (27%).

The sensitivity analysis regarding the imputation is shown in S3 Table For the range of δ-values chosen, there was a marginally/small increase in the estimated coefficient of albuminuria in HUNT2, including the p-values. The sensitivity analysis confirms that the analysis is robust to alternative plausible missing data assumptions, under all specified MAR mechanisms of missingness. In addition, the MI estimates are close to those from the complete case analysis.

## Discussion

To our knowledge, this is the first study to examine the bidirectional relationship between depression symptoms and future albuminuria and simultaneously assess the relationship between albuminuria and future depression symptoms. In this 10-year follow-up study, there was no association between depression symptoms, measured by HADS score, and future albuminuria, nor between albuminuria and future depression symptoms.

Even though albuminuria in HUNT2 was significantly associated with HADS-D symptoms in HUNT3, the correlation coefficient of 0.07 indicates a negligible association between the two. This corresponds to albuminuria explaining only 0.007% of the variance in depression

**Table 3. General linear regression[1] of the effects of albuminuria in HUNT2 on depression symptoms in HUNT3.**

| Variable | HADS-Depression 7-items HUNT3 (+1)[2] | | | HADS-Depression 4-items[3] HUNT3 (+1)[2] | | |
|---|---|---|---|---|---|---|
| | Model 1 | Model 2 | Model 3 | Model 1 | Model 2 | Model 3 |
| | β (95% CI) | β (95% CI) | β (95% CI) | β (95% CI) | β (95% CI) | β (95% CI) |
| Intercept[4] | 2.60 (2.10,3.09)*** | 1.73 (1.35,2.11)*** | .61 (-.90,2.13) | 2.03 (1.72,2.34)*** | 1.67 (1.41,1.92)*** | 1.05 (.04,2.08)* |
| Albuminuria HUNT2[5] | .006 (-.02,.03)[5] | .006 (-.02,.03)[5] | -.001 (-.02,.02)[5] | .0004 (-.01,.02) | .0002 (-.01,.01) | -.003 (-.01,.01) |
| Age (years) | .04 (.03,.05)*** | .02 (.01,.03)*** | .01 (.003,.03)* | .02 (.02,.03)*** | .01 (.01,.02)*** | .009 (.001,.02)* |
| Male sex | .34 (.11,.56)** | .12 (-.06,.30) | -.004 (-.31,.30) | -.15 (-.28,-.01)* | -.19 (-.30,-.08)* | -.25 (-.45,-.05)* |
| HADS score HUNT2[6] | | .54 (.50,.58)*** | .53 (.49,.57) *** | | .49 (.45,.53)*** | .49 (.45,.53)*** |
| Education or work, Intermediate | | | -.26 (-.47,-.04)* | | | -.11 (-.25,.03) |
| Education or work, Salariat | | | -.30 (-0.56,-.04)* | | | -.09 (-.25,.08) |
| Body Mass Index (kg/m²) | | | .04 (.01,.06)** | | | .02 (.01,.04) ** |
| Smoking (Previous) | | | -.002 (-.20,.20) | | | -.06 (-.21,.07) |
| Smoking (Current ) | | | .53 (.23,.82)*** | | | .24 (.07,.41)** |
| Cholesterol (mg/mmol) | | | .01 (-.07,.09) | | | -.0002 (-.05,0.05) |
| Antihypertensive medication | | | .19 (-.02,.40) | | | .12 (-.01,.26) |
| CVD (yes)[7] | | | .23 (-.09,.55) | | | .12 (-.09,.34) |
| eGFR (CKD-epi) | | | .01 (-.002,.02) | | | .004 (-.003,.01) |
| Diabetes (yes) | | | .33 (.04,.64)* | | | .23 (.04,.42)* |
| Systolic blood pressure | | | -.002 (-.01,.003) | | | -.002 (-.005,.001) |
| 1–7 units per week alcohol | | | -.22 (-.42,-.02)* | | | -.05 (-.18,.08) |
| 8–14 units per week | | | -.43 (-.74,-.12)** | | | -.09 (-.29,.11) |
| >15 units per week | | | -.35 (-.89,.20) | | | -.19 (-.54,.16) |

The grey area in the table indicates the results for the covariates (*k*) in the model and is not the focus of this article.

[1] Equation 1 explaining the GLM approach for depression: $\gamma_i = \beta_0 + [\![\beta_{Sex} X]\!]\_{(i,Sex)} + [\![\beta_{Age} X]\!]\_{(i,Age)} + \cdots + [\![\ [\![\beta_k X]\!]\_{(i,k)} + \varepsilon]\!]\_i$ where i is the individual participant, γi is the HADS score given in HUNT3 for individual i, β0 is the intercept, and *k* = confounder variables and εi is the residual effect.

[2] For Equation 1 a value of +1 was added to the score as gamma distribution does not exist for 0, and HADS scores include the value 0. When transforming back to the original scale, we find that the expectation (E) of model 1–3 is E(y+1) = E(y) + 1. Thereby, the only transformation is for the intercept (b0–1).

[3] HADS-D 4-item is the first 4 items of the Hospital Anxiety and Depression Score-1111Depression subscale

[4] The intercept was back-transformed -1.

[5] Albuminuria was measured as Albumin Creatinine ratio (ACR) in urine and treated as a continuous variable.

[5] The β indicates points change in depression score in HUNT 3 associated with one unit increase in albuminuria value at HUNT2.

[6] Depression was measured by the Hospital anxiety and Depression scale and treated as a continuous variable.

[7] CVD = self-reported cardiovascular disease

Model 1 included *k* = age and sex. Model 2 included *k* = Model 1+baseline level of the outcome variable. Model 3 included *k* = education, body mass index (BMI), smoking status, cholesterol, eGFR, diabetes, systolic blood pressure, blood pressure medication and alcohol in addition to model 2.

All the predictors, *k*, in the model were measured at HUNT2.

*Denotes significance p<0.05.

** Denotes significance at p<0.01.

*** Denotes significance at p < .001.

levels in HUNT3. Thus, albuminuria is probably not a useful biomarker for future depression symptoms. HADS-D in HUNT2 had no significant correlation with albuminuria in HUNT3, with a correlation coefficient of 0.04 for HADS-D7 and 0.03 for HADS-D4. With wide confidence intervals including the null, 0.04% of the variance in HUNT 3 albuminuria levels was explained by previous depression levels, making depression symptoms as measured by HADS ineffectual as a biomarker for albuminuria.

The only previous longitudinal study of future depression symptoms found a HR 1.25 (95% CI, 0.94 to 1.65) for depression with albuminuria ≥30 mg/g compared to those with

**Table 4. General linear regression[1] of the effects of depression symptoms in HUNT2 on albuminuria in HUNT3.**

| Variable | Model 1 (log)[3] β (95% CI) | Model 2 (log)[3] β (95% CI) | Model 3 (log)[3] β (95% CI) | Model 3[3] % Change (95 CI) | Model 3[4] (log) β (95% CI) | Model 3[4] % change (95% CI) |
|---|---|---|---|---|---|---|
| | Albuminuria[2] levels in HUNT3 (log and %change) | | | | | |
| Intercept | -.90 (-1.13,-.66)*** | -.90 (-1.13,-.67)*** | -1.85 (-2.63,-1.07)*** | | -1.77 (-2.52, -1.02)*** | |
| HADS-D score HUNT2[5] | .006 (-.01,.02)[4] | .005 (-.01,.02)[4] | .005 (-.01,.02)[4] | 0.50 (-0.99, 2.02) | -.004 (-.03, .02)[5] | -0.40 (-2.96, 2.02) |
| Age (years) | .02 (.02,.03)*** | .02 (.02,.03)*** | .02 (.02,.03)*** | | .02 (.02, .03)*** | |
| Male sex | .13 (.04,.22)** | .12 (.03,.21)** | -0.03 (-.18,.12) | | -.02 (-.17, .13) | |
| Albuminuria HUNT2 | | .04 (.03,.05)*** | .04 (.03,.05)*** | 4.08 (3.05, 5.13)*** | .04 (.03,.05)*** | 4.08 (3.05, 5.13)*** |
| Education (reference) | | | | | | |
| 10–12 years | | | .08 (.-.01,.19) | | .08 (-.03,.18) | |
| >12 years | | | .11 (-.01,.24) | | .09 (-.04,.22) | |
| Body Mass Index (kg/m[2]) | | | .003 (-.01,.01) | | .002 (-.01,.01) | |
| Smoking (reference) | | | | | | |
| Previous | | | .04 (-.07,.14) | | .03 (-.07,.14) | |
| Current | | | .22 (.10,.36)*** | | .22 (.10,.34)*** | |
| Cholesterol (mg/mmol) | | | -0.02 (-.06,.02) | | -.01 (-.05,.03) | |
| BP medication (yes) | | | .15 (.04,.27)** | | .15 (.06,.25)** | |
| CVD (yes)[5] | | | .001 (-.14,.14) | | .02 (-.12,.16) | |
| eGFR (CKD-epi) | | | .005 (-.001,.01) | | .005 (-.001,.01) | |
| Diabetes (yes) | | | .35 (.18,.51)*** | | .32 (.18,.47)*** | |
| Systolic blood pressure (mmHg) | | | .002 (-.001,.005) | | .002 (-.001,.004) | |
| Alcohol (reference) | | | | | | |
| 1–7 units per week | | | .02 (-.08,.11) | | .02 (-.08,.11) | |
| 8–14 units per week | | | -.02 (-.19,.15) | | -.004 (-.17,.17) | |
| 15 units per week | | | -.05 (-.35,.26) | | -.06 (-.37,.25) | |

[1] Equation 2 explaining the GLM approach for albuminuria: $z_i = \beta_0 + [\![\beta_{Sex} X]\!]\_{iSex} + [\![\beta_{Age} X]\!]\_{iAge} + \cdots + [\![\, [\![\beta_k X]\!]\_{(i,k)} + \varepsilon]\!]\_i$ where zi is measured albuminuria level for individual i in HUNT3.

[2] Albuminuria is measured as Albumin Creatinine ratio (ACR) in urine and treated as a continuous variable.

[3] Depression is measured by the 7 item depression subscale of the Hospital Anxiety and Depression Scale (HADS-D). The score is used as a continuous variable.

[4] Depression is measured by the 4 first items of the HADS-D. The score is used as a continuous variable.

[5] CVD = self-reported cardiovascular disease

The β indicates change in albuminuria levels in HUNT 3 associated with one unit increase in HADS-D score at HUNT2.

Model 1 included k = age and sex.

Model 2 included k = Model 1+baseline level of the outcome variable.

Model 3 included k = education, body mass index, smoking status, cholesterol, eGFR, diabetes, systolic blood pressure, blood pressure medication and alcohol in addition to model 2. All the predictors, k, in the model were measured at HUNT2.

The grey area in the table indicates the results for the covariates (k) in the model and is not the focus of this article.

*Denotes significance p<0.05.

** Denotes significance at p<0.01.

*** Denotes significance at p < .001.

95% confidence intervals (CI) for the regression coefficient β are reported in parenthesis

albuminuria <10 mg/g [9]. Thus, the wide and inconclusive confidence intervals is concordant with our study, in that albuminuria is not a reliable biomarker of future depression. Liu et al. (2022) found that eGFR was a more robust marker for future depression, and several cross sectional studies also support eGFR as a possible marker for depression [2, 32, 33]. However, we did not aim to assess eGFR in our study. No previous studies have assessed if depression symptoms are a useful biomarker for future albuminuria, and thus we provide the first longitudinal results suggesting the failure of HADS-D symptoms to predict albuminuria.

While Liu (2022) et. als [9] findings of a longitudinal relationship between albuminuria and future depression were not significant, their point estimate suggested a possible risk for depression with albuminuria. This difference between our study and Liu et. al (2022) could be explained by the latter using Center for Epidemiologic Studies Depression Scale (CES-D), a scale that is broadly comparable and equivalent to the HADS-D scale, except that the ability to detect major depression is better with the CES-D than HADS-D [34]. Limitations of our study include the use of self-reported depression measures. One previous cross-sectional study which used the MINI-International Neuropsychiatric interview [35] found that the odds ratio for a minor or major depressive episode was 2.13 (95% CI 1.36–3.36) for albuminuria 15–<30 mg/24 h. The HADS questionnaire is a well-validated screening instrument for anhedonia and psychological distress [26], and performs well in screening the general population and is one of the most commonly used screening tools for depression in medically ill populations. The lack of structured clinical interviews, unlike for example the MINI-International Neuropsychiatric interview used by Martens et al [1] in confirming a clinical diagnosis of depression could be a limitation in our study, especially due to the absence of sleep items and any other vegetative items. The Structured Clinical Interview for DSM-5 (SCID-5) and Mini-International Neuropsychiatric interview are generally referred to as the gold standard for depression assessments, however the HADS-D is shown acceptable against this gold standard (AUC 0.83) [36]. The HADS-D scale has shown a cross-sectional association between depression and albuminuria, however the results were explained by confounding by age and sex [3]. A further limitation is that the association between somatic depressions symptoms and full diagnostic depression is not measured in this study. However, we added analyses of the shorter HADS-4 item version, due to psychometric analyses of the HADS that have shown less heterogeneous phenotype measures for HADS-4 in genetic analyses [27], but this did not change the results.

Our study design was able to detect that albuminuria levels at HUNT3 was explained by 4% of the albuminuria levels in HUNT2, and 27% of depression levels in HUNT3 were explained by depression levels in HUNT2. Thus, if albuminuria and the HADS-D symptom levels were strongly linked, our design should be able to detect that, although we acknowledge that more repeated measurements might have strengthened our design.

A strength of this study was the measurement of albuminuria in three urine samples [21, 37, 38]. Using the same urine sampling method, one previous cross-sectional study from the HUNT survey found that age and comorbidity explained the association between albuminuria and depression symptoms [3]. All these HUNT studies give greater precision in albuminuria analysis compared to results from one spot urine sample [19] as used in the cross-sectional study that found an association between albuminuria and depression in a diabetic sample by Yu et al (2013) [2, 39]. Further, our use of fresh urine samples avoided problems with frozen samples [19]. Fresh urine samples are shown as valid as 24 hours urine sampling [40] to establish albuminuria. One cross-sectional study used 24 hours of urine sampling and found an association between depression and albuminuria [1].

## Conclusion

The gap in life expectancy seen between persons with and without mental health problems largely stem from physical disease [41]. It is important to detect and manage albuminuria in persons with and without depression symptoms to prevent kidney disease [42] and cardiovascular disease [43], as well as manage common risk factors for both.

In this first follow-up study to firstly assess the association between depression symptoms and future albuminuria and secondly assess the ability of albuminuria to predict depression symptoms, there was no evidence to support a bidirectional causal relationship between these two

conditions. Future studies should investigate whether other depression measurements yields different results and whether eGFR is a better biomarker for future depression than albuminuria.

## Supporting information

**S1 File. Overview of statistical models.**
(DOCX)

**S1 Table. General linear regression results in complete data set.**
(DOCX)

**S2 Table. Analysis of Variance (ANOVA) for the covariates in HUNT2 explaining albuminuria or depression in HUNT3.**
(DOCX)

**S3 Table. Sensitivity analysis for the imputed data for model 3 with depression for HUNT 3 as response.**
(DOCX)

## Acknowledgments

The Nord-Trøndelag Health Study (the HUNT Study) is a collaboration between the HUNT Research Centre (Faculty of Medicine and Health Sciences, Norwegian University of Science and Technology, NTNU), Nord-Trøndelag County Council, Central Norway Regional Health Authority, and the Norwegian Institute of Public Health.

## Author Contributions

**Conceptualization:** Lise Tuset Gustad, Torfinn Hynnekleiv, Ottar Bjerkeset, Solfrid Romundstad.

**Data curation:** Lise Tuset Gustad, Anna Marie Holand, Ottar Bjerkeset, Solfrid Romundstad.

**Formal analysis:** Lise Tuset Gustad, Anna Marie Holand.

**Funding acquisition:** Lise Tuset Gustad, Ottar Bjerkeset, Solfrid Romundstad.

**Investigation:** Lise Tuset Gustad, Anna Marie Holand, Solfrid Romundstad.

**Methodology:** Lise Tuset Gustad, Anna Marie Holand, Torfinn Hynnekleiv, Solfrid Romundstad.

**Project administration:** Lise Tuset Gustad, Ottar Bjerkeset.

**Resources:** Lise Tuset Gustad.

**Software:** Lise Tuset Gustad, Anna Marie Holand.

**Supervision:** Solfrid Romundstad.

**Writing – original draft:** Lise Tuset Gustad, Anna Marie Holand.

**Writing – review & editing:** Torfinn Hynnekleiv, Ottar Bjerkeset, Michael Berk, Solfrid Romundstad.

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
