## [Decision Letter · Decision Letter 0]

23 Jun 2022

PONE-D-22-08418A bidirectional association study between depressive symptoms and albuminuria – A longitudinal population-based cohort with repeated measures from the HUNT2 and HUNT3 StudyPLOS ONE

Dear Dr. Gustad,

Thank you for submitting your manuscript to PLOS ONE. After careful consideration, we feel that it has merit but does not fully meet PLOS ONE’s publication criteria as it currently stands. Therefore, we invite you to submit a revised version of the manuscript that addresses the points raised during the review process.

We look forward to receiving your revised manuscript.

Kind regards,

Angela Lupattelli, PhD

Academic Editor

PLOS ONE

Journal Requirements:

“MB has received Grant/Research Support from the NIH, Cooperative Research Centre, Simons Autism Foundation, Cancer Council of Victoria, Stanley Medical Research Foundation, Medical Benefits Fund, National Health and Medical Research Council, Medical Research Futures Fund, Beyond Blue, Rotary Health, A2 milk company, Meat and Livestock Board, Woolworths, Avant and the Harry Windsor Foundation, has been a speaker for Astra Zeneca, Lundbeck, Merck, Pfizer, and served as a consultant to Allergan, Astra Zeneca, Bioadvantex, Bionomics, Collaborative Medicinal Development, Lundbeck Merck, Pfizer  and Servier – all unrelated to this work.”

Reviewers' comments:

Reviewer's Responses to Questions

**Comments to the Author**

1. Is the manuscript technically sound, and do the data support the conclusions?

Reviewer #1: Yes

Reviewer #2: Yes

2. Has the statistical analysis been performed appropriately and rigorously? 

Reviewer #1: Yes

Reviewer #2: Yes

3. Have the authors made all data underlying the findings in their manuscript fully available?

Reviewer #1: Yes

Reviewer #2: Yes

4. Is the manuscript presented in an intelligible fashion and written in standard English?

Reviewer #1: Yes

Reviewer #2: Yes

5. Review Comments to the Author

Reviewer #1: This is an interesting paper, in which the bidirectional relation between albuminuria and depressive symptoms as measured by HADS was investigated in a longitudinal way. Unlike some cross sectional studies, no clinically relevant association was observed.

The study is well performed and the paper is well written. I have no major remarks, but given the discrepancy between studies that used different ways to assess depressive symptoms (e.g. HADS and mini International Neuropsychiatric Interviews. To which extent do these scales measure the same or different constructs? This would be relevant to know for the reader in order to interprete these discrepancies.

In the title "depressive symptoms by HADS"" might be the most appropriate term

Reviewer #2: A BIDIRECTIONAL ASSOCIATION STUDY BETWEEN DEPRESSIVE SYMPTOMS AND ALBUMINURIA – A LONGITUDINAL POPULATION-BASED COHORT WITH REPEATED MEASURES FROM THE HUNT2 AND HUNT3 STUDY

General comments

This paper presents findings on important areas such as depression, albuminuria, and CVD. The use of cohort compared to cross-sectional studies helps improve the scant knowledge available on the hypothesis generated in the study. The use of robust statistical tools is commendable.

The article will benefit from proof reading to correct a few grammar and typographical errors.

Introduction

The introductory section will benefit from clearly defining the purpose of the study. Authors begin by linking CVD to depression and albuminuria which is fine. Then this is lost and a focus is now on the bidirectional relationship between depression and albuminuria.

I do understand that the article hinges on the bidirectional relationship between albuminuria and depression and this was emphasized in paragraphs 2 and 3 of the introductory section.

In the second paragraph, authors stressed on the fact that only one study has used a cohort, then they introduced a cross-sectional cohort study in the third paragraph. They also make the point that this is the first cohort study looking at the bidirectional relationship between depression and albuminuria. I suggest, authors clarify this and/or make this important point clear as it provides a good strength for the paper.

Methods

I have no comment on this section

Results

Authors report no significant association between depression levels and albuminuria and vice versa but report a that some small percentage of the relationship is also attributable to depression and to albuminuria. This sounds confusing, can authors clarify these statements.

Table 2 should be improved by creating a heading to show what r, what is in the bracket and the p value stands for or can be explained at the foot note.

Discussion

Again, authors repeat this statement “For albuminuria, only 0.04% is explained by previous depression levels, and for depression symptoms, only 0.007% is explained by previous albuminuria levels”. Ideally, there should first be an association and then we go and look for the “weight” influence of the explanatory variables.

Comparison of studies which used proteinuria to current study which used albuminuria is a little out of place. Especially, when authors have hammered on its limitations in comparison to albuminuria and that is the more reason why they are using albuminuria. Is it possible to find more suitable articles to support their discussions?

The last sentence in paragraph 3 is incomplete. Authors should kindly resolve this since it was addressing an important point

Conclusion

In the conclusion, authors report no association between depression levels and albuminuria and vice versa. However, authors failed to provide guidance on the next action or way forward.

6. PLOS authors have the option to publish the peer review history of their article (what does this mean?). If published, this will include your full peer review and any attached files.

Reviewer #1: No

Reviewer #2: **Yes: **David Nana Adjei

---

## [Author Response · Author response to Decision Letter 0]

14 Aug 2022

We would like to thank the editor and the reviewers for putting in time and effort to elevate the quality of our manuscript. All editorial concerns are answered in the cover letter and applicable changes are done in manuscript. All reviewers concerns have been addressed and revisions to the manuscripts have been applied accordingly. 

Point-by-point responses to editor are provided in the cover letter. Point-by-point responses to reviewers are provided below, and additionally we attach a rebuttal letter, which details where in the manuscript changes are made. We have provided a copy with track changes and a clean version of the manuscript. 

6. Review Comments to the Author

Reviewer #1: This is an interesting paper, in which the bidirectional relation between albuminuria and depressive symptoms as measured by HADS was investigated in a longitudinal way. Unlike some cross sectional studies, no clinically relevant association was observed.

The study is well performed and the paper is well written. I have no major remarks, but given the discrepancy between studies that used different ways to assess depressive symptoms (e.g. HADS and mini International Neuropsychiatric Interviews. To which extent do these scales measure the same or different constructs? This would be relevant to know for the reader in order to interprete these discrepancies.

Authors response: In the discussion we now incorporate more detailed discussions about the differences between studies that might explain the conflicting results. 

In the title "depressive symptoms by HADS"" might be the most appropriate term

Authors response: We changed the title according to reviewers suggestions and the new title reads; 

“The bidirectional association depressive symptoms, assessed by the HADS and albuminuria – A longitudinal population-based cohort study with repeated measures from the HUNT2 and HUNT3 Study” (Title)

Reviewer #2: A BIDIRECTIONAL ASSOCIATION STUDY BETWEEN DEPRESSIVE SYMPTOMS AND ALBUMINURIA – A LONGITUDINAL POPULATION-BASED COHORT WITH REPEATED MEASURES FROM THE HUNT2 AND HUNT3 STUDY

General comments

This paper presents findings on important areas such as depression, albuminuria, and CVD. The use of cohort compared to cross-sectional studies helps improve the scant knowledge available on the hypothesis generated in the study. The use of robust statistical tools is commendable.

The article will benefit from proof reading to correct a few grammar and typographical errors.

Authors response: Careful proofreading is now performed by all co-authors and in particular professor Michael Birk that has English as a first language. We added this in the authors´ contribution: 

“All authors interpreted the results and critically edited and proofread the manuscript ensuring intellectual content of critical importance to the work described. MB did the final English proofreading. All authors approved the final version of the manuscript” (authors´ contribution page 30)

Introduction

The introductory section will benefit from clearly defining the purpose of the study. Authors begin by linking CVD to depression and albuminuria which is fine. Then this is lost and a focus is now on the bidirectional relationship between depression and albuminuria.

I do understand that the article hinges on the bidirectional relationship between albuminuria and depression and this was emphasized in paragraphs 2 and 3 of the introductory section.

Authors response: We have reorganized the introductory section in order to make our aim more clear. 

In the second paragraph, authors stressed on the fact that only one study has used a cohort, then they introduced a cross-sectional cohort study in the third paragraph. They also make the point that this is the first cohort study looking at the bidirectional relationship between depression and albuminuria. I suggest, authors clarify this and/or make this important point clear as it provides a good strength for the paper.

Authors response: We understand that we have not been clear about cross-sectional and longitudinal studies. We hope that the reorganization of the introduction described above is satisfactory in solving this issue as well. Since we sent in our manuscript one new longitudinal study have been published that has measured albuminuria and depression repeatedly; published in 2022 by Liu et al [1], which we have included in the manuscript. That paper used the Center for Epidemiologic Studies Depression Scale (-CES-D),and did not, like us, find a convincing longitudinal relation between albuminuria and depression. The HR (95% confidence interval) for depression with UACR ≥30 mg/g compared to those with UACR <10 mg/g was 1.25 (95% CI, 0.94 to 1.65). Our study thus confirm the uncertainty related to albuminuria being a useful biomarker for future depression and still our study is the first study to assess the usefulness to use albuminuria as a biomarker for depression. 

… 

Methods

I have no comment on this section

Results

Authors report no significant association between depression levels and albuminuria and vice versa but report a that some small percentage of the relationship is also attributable to depression and to albuminuria. This sounds confusing, can authors clarify these statements.

Authors respons; We agree that this was confusing and have now altered the statements. We hope the new statement is better understood. The paragraph now reads; 

“Even though albuminuria in HUNT2 was significantly associated with HADS-depression symptoms in HUNT3, the correlation coefficient of 0.07 indicates a negligible association between the two. This corresponds to albuminuria explaining only 0.007% of the variance in depression levels in HUNT3. Thus albuminuria is probably not a useful biomarker for future depression symptoms. HADS-depression in HUNT2 had no significant correlation with albuminuria in HUNT3, with a correlation coefficient of 0.04 for HADS7 and 0.03 for HADS4. With wide confidence intervals including the null, 0.04% of the variance in HUNT 3 albuminuria levels was explained by previous depression levels, making depression symptoms as measured by HADS ineffectual as a biomarker for albuminuria..” Discussion page 24. 

Authors reply; 

Table 2 should be improved by creating a heading to show what r, what is in the bracket and the p value stands for or can be explained at the foot note.

Authors response: We have rearranged our table footnotes in order to put the explanations of our abbreviations more central 

“Abbreviations: r =correlation coefficient, ()=(standard error of pooled estimates), p= p-value.” (Table 2)

Discussion

Again, authors repeat this statement “For albuminuria, only 0.04% is explained by previous depression levels, and for depression symptoms, only 0.007% is explained by previous albuminuria levels”. Ideally, there should first be an association and then we go and look for the “weight” influence of the explanatory variables.

Authors reply: We have rephrased this part of the discussion as stated above on page 6 in this rebuttal letter. 

Comparison of studies which used proteinuria to current study which used albuminuria is a little out of place. Especially, when authors have hammered on its limitations in comparison to albuminuria and that is the more reason why they are using albuminuria. Is it possible to find more suitable articles to support their discussions?

Authors responce: Since we sent in our manuscript one new longitudinal study have been published that has measured albuminuria and depression repeatedly; published in 2022 by Liu et al [1], we have now discussed our findings against that paper instead-

The last sentence in paragraph 3 is incomplete. Authors should kindly resolve this since it was addressing an important point

Authors response: Thank you for spotting this sentence. We have removed one word that caused the unclarity of the sentence. The new sentence reads;

“One cross-sectional study used 24 hours urinary sampling in finding an association between depression and albuminuria” (Discussion page 27)

Conclusion

In the conclusion, authors report no association between depression levels and albuminuria and vice versa. However, authors failed to provide guidance on the next action or way forward.

Authors response: We have added the future way forward in the conclusion: 

“Future studies should investigate whether other depression measurements yields different results and whether eGFR is a better biomarker for future depression than albuminuria.” (Conclusion, page 28)

6. PLOS authors have the option to publish the peer review history of their article (what does this mean?). If published, this will include your full peer review and any attached files.

Do you want your identity to be public for this peer review? For information about this choice, including consent withdrawal, please see our Privacy Policy.

Reviewer #1: No

Reviewer #2: Yes: David Nana Adjei

---

## [Editor Report · Decision Letter 1]

25 Aug 2022

The bidirectional association between depressive symptoms, assessed by the HADS, and albuminuria – A longitudinal population-based cohort study with repeated measures from the HUNT2 and HUNT3 Study

PONE-D-22-08418R1

Dear Dr. Gustad,

We’re pleased to inform you that your manuscript has been judged scientifically suitable for publication and will be formally accepted for publication once it meets all outstanding technical requirements.

Kind regards,

Angela Lupattelli, PhD

Academic Editor

PLOS ONE

---

## [Editor Report · Acceptance letter]

1 Sep 2022

PONE-D-22-08418R1 

The bidirectional association between depressive symptoms, assessed by the HADS, and albuminuria – A longitudinal population-based cohort study with repeated measures from the HUNT2 and HUNT3 Study 

Dear Dr. Gustad:

I'm pleased to inform you that your manuscript has been deemed suitable for publication in PLOS ONE. Congratulations! Your manuscript is now with our production department. 

Kind regards, 

on behalf of

Dr. Angela Lupattelli 

Academic Editor

PLOS ONE